# Different Impacts of Cryopreservation in Endothelial and Epithelial Ovarian Cells

**DOI:** 10.3390/ijms241512225

**Published:** 2023-07-31

**Authors:** Julian Marschalek, Marlene Hager, Sophie Wanderer, Johannes Ott, Maria Frank, Christian Schneeberger, Detlef Pietrowski

**Affiliations:** 1Department of Obstetrics and Gynecology, Medical University of Vienna, Spitalgasse 23, 1090 Vienna, Austria; julian.marschalek@meduniwien.ac.at (J.M.); marlene.hager@meduniwien.ac.at (M.H.); johannes.ott@meduniwien.ac.at (J.O.); maria.frank@meduniwien.ac.at (M.F.); christian.schneeberger@meduniwien.ac.at (C.S.); 2FH Campus Wien, University of Applied Science, 1100 Vienna, Austria

**Keywords:** cryopreservation, vitrification, dimethyl sulfoxide (DMSO), cryopreservation-induced delayed-onset cell death, granulosa cell

## Abstract

The aim of our laboratory-based study was to investigate the extent of delayed-onset cell death after cryopreservation in endothelial and epithelial cell lines of ovarian origin. We found differences in percentages of vital cells directly after warming and after cultivation for 48 to 72 h. A granulosa cell line of endothelial origin (KGN) and an epithelial cell line (OvCar-3) were used. In both DMSO-containing and DMSO-free protocols, significant differences in vitality rates between the different cell lines when using open and closed vitrification could be shown (DMSO-containing: KGN open vs. OvCar open, *p* = 0.001; KGN closed vs. OvCar closed, *p* = 0.001; DMSO-free: KGN open vs. OvCar open, *p* = 0.001; KGN closed vs. OvCar closed, *p* = 0.031). Furthermore, there was a marked difference in the percentage of vital cells immediately after warming and after cultivation for 48 to 72 h; whereas the KGN cell line showed a loss of cell viability of 41% using a DMSO-containing protocol, the OvCar-3 cell loss was only 11% after cultivation. Using a DMSO-free protocol, the percentages of late-onset cell death were 77% and 48% for KGN and OvCar-3 cells, respectively. Our data support the hypothesis that cryopreservation-induced damage is cell type and cryoprotective agent dependent.

## 1. Introduction

Cryopreservation of ovarian tissue is an increasingly applied method for fertility preservation in patients undergoing gonadotoxic therapies, and has led to the birth of more than 200 children so far [1,2]. In general, there are slow and fast freezing methods. Vitrification, an ultra-fast freezing method, is promising, and probably cost-effective [3]. The process of cryopreservation is complex, and, especially in basic research, the evaluation of its success is usually based on sample assessment immediately after warming [4,5,6].

When storing biological living cells and tissues at cryogenic temperatures, cells are exposed to physical and chemical stressors, such as cold and osmotic stress, or the formation of intracellular and extracellular ice crystals. Regardless of advancements in cryopreservation protocols, the potential toxicity of cryoprotective agents is also reported to play a role in the development of cryopreservation-induced injuries. As most cryoinjuries occur during freezing and warming processes, cell death after warming is still substantial, and occurs mostly within the first 24 h of culturing [7]. 

Furthermore, cryopreservation-induced delayed-onset cell death (CIDOCD) is a regularly observed phenomenon in cryopreservation, which occurs at a molecular level via apoptotic and necrotic processes, and hours after an apoptosis-triggering event. Cell loss is reported to be as high as 70% of cells when examined 24–48 h after warming [4,8]. A number of studies have shown that CIDOCD is not only cell-type specific, but also dependent upon the growth phase of the respective cells and the cryoprotective agent used [9,10]. 

Thus, evaluating the effectivity of a cryopreservation via sample assessment immediately after warming only has informative value to a limited extent [4].

As stated above, an upcoming and alternative method of cryopreserving ovarian tissue is vitrification, an ultra-fast freezing method that prevents cell-damaging ice crystal formation. Despite promising research results in human and animal studies indicating similar outcomes compared to the conventional slow freezing method, there are no established standard vitrification protocols yet [11]. The most significant differences in published protocols are the different CPAs used and the times required for respective equilibration steps [12,13]. The most popular CPAs are dimethyl sulfoxide (DMSO), glycerol, propanediol (PrOH), and ethylene glycol (EG), which are necessary to achieve controlled cell dehydration before cryopreservation. For this step, the mentioned permeating agents are used together with non-permeating agents, which include proteins, sugars, and other macromolecules [14]. Notably, CPAs themselves can have a dose-dependent toxic effect on cells and their functions [15,16].

In this regard, it is still unclear whether dimethyl sulfoxide (DMSO), which is used in many protocols, is the most appropriate CPA for human reproductive tissue [17,18]. Moreover, it is still uncertain if open or closed vitrification devices result in better outcomes in oocyte, embryo, and ovarian tissue vitrification [19,20]. 

It must be taken into account that ovarian tissue consists of a multitude of different cells and cell types, mostly of epithelial and endothelial origin, of which the granulosa cells surrounding the follicle are particularly important for ovarian function. Of note, the individual effects of vitrification on a particular cell type have not been sufficiently investigated up to now. A few questions remain to be answered, including if the use of open or closed vitrification devices makes a difference concerning the vitrification of specific cell types, and whether late-onset cell death can also occur in vitrified ovarian tissue. If this is the case, it would be interesting to know which cell type is the most affected [19,21]. 

Thus, the aim of our study was to investigate the effects of open and closed vitrification devices with respect to vitality in endothelial and epithelial cell lines, using a DMSO-containing and a DMSO-free protocol. 

## 2. Results

### 2.1. FACS Analysis 

After vitrification using the DMSO-containing protocol and subsequent warming (Figure 1), 72.79% of KGN cells (KGN O) were vital using an open vitrification device. After closed vitrification, 74.56% of KGN cells (KGN CL) were vital (KGN O vs. KGN CL *p* = 0.899). There were 52.66% vital OvCar-3 cells (OvCar O) after open vitrification, and 39.1% vital OvCar-3 cells (OvCar CL) after closed vitrification (OvCar O vs. OvCar CL *p* = 0.0746). 

A significant difference was found in the amount of vital cells resulting from open and closed vitrification with respect to the different cells used (KGN O vs. OvCar O *p* = 0.001; KGN CL vs. OvCar CL *p* = 0.001).

Using the DMSO-free vitrification protocol (Figure 2), 85.18% of KGN cells (KGN O) were vital using an open vitrification device, whereas 78.41% of KGN cells (KGN CL) were vital after closed vitrification (KGN O vs. KGN CL *p* = 0.828). The percentages of vital OvCar-3 cells were 63.57% (OvCar O) and 63.32% (OvCar CL) (OvCar O vs. OvCar CL *p* = 0.899).

There was a significant difference in KGN and OvCar cell vitality after open and closed vitrification (KGN O vs. OvCar O, *p* = 0.001; KGN CL vs OvCar CL, *p* = 0.031).

### 2.2. Cell Cultivation for 48–72 h and Trypan Blue Staining

Trypan blue staining results are presented in Figure 3 and Figure 4. With the DMSO-containing protocol warming and cultivation, 29.09% of KGN cells were vital using an open vitrification device (KGN O). After closed vitrification, 35.83% of KGN cells were vital (KGN CL). The percentages of vital OvCar-3 cells after cultivation were 42.77% (OvCar O) and 27.71% (OvCar CL).

After DMSO-free vitrification, warming, and cultivation, 5.41% of KGN cells were vital using an open vitrification device (KGN O). After closed vitrification, 4.54% of KGN cells were vital (KGN CL). When comparing open and closed vitrification methods, 16.54% (OvCar O) and 15.31% (OvCar CL) of OvCar-3 cells were vital, respectively. 

### 2.3. Total Cell-Specific Vitality Rate

The vitality rates of non-vital cells after vitrification and cultivation are presented in Table 1. They indicate that, using the DMSO-containing protocol, the percentage of non-vital cells after cultivation was lower in both KGN and OvCar-3 cell lines compared to using the DMSO-free protocol (41.22% and 10.64% vs. 76.82% and 47.52%, respectively). Comparing the percentages of non-vital cells in the open protocol and closed protocol, no clear differences were apparent (61.74% and 28.46% vs. 56.3% and 29.7%, respectively).

## 3. Discussion

In this study, we investigated and quantified the extent of delayed-onset cell death after cryopreservation using vitrification in two different cell lines of ovarian origin. 

The main finding of our study was a marked difference in the percentage of vital cells immediately after warming and after cultivation for 48 to 72 h in a cell-type specific manner. Whereas the KGN cell line showed a loss of cell viability of 41% using a DMSO-containing protocol, the OvCar-3 cell line seemed to be more resistant, as its cell loss was only 11% after cultivation. Using a DMSO-free protocol, the percentages of late-onset cell death were 77% and 48% for KGN and OvCar-3 cells, respectively.

After the warming process in cell cryopreservation, a low recovery rate of cells [4,22] or loss of proper cell function [23,24] are typical findings. Whereas a low recovery rate of cells directly after warming is mainly linked to cellular damage by intra-cellular ice crystal formation, osmotic pressure, apoptosis, or toxicity, the phenomenon of late-onset cell death after cryopreservation is different, and was first described by Baust and colleagues [8]. CIDCOD has been reported to occur later than 24 h after warming, and is primarily induced by apoptotic and/or necrotic processes.

The apoptosis-triggering processes of CIDCOD are reported to involve a genomic response via the up-regulation of key apoptotic enzymes of the caspase family [4] and increased caspase-3 and caspase-6 activity [25]. Moreover, decreased cytokine production and reduced lymphocyte growth were observed 72 h after cultivation in repeated cryopreservation/warming cycles. [6]. Our study group noted that the expression of tumor necrosis factor alpha (TNF-alpha), which is a key factor of late apoptosis modulating inflammatory reaction, was significantly increased after vitrification and subsequent warming in KGN cell lines [26].

Our results demonstrate a clear difference between the vitality of cells immediately after warming and at a later time point, supporting the involvement of CIDCOD. Of note, most articles regarding cell cryopreservation evaluate cell vitality solely directly after warming. 

Another interesting observation from our study was the different sensitivities of specific cell types to CIPCOD. Whereas the endothelial granulosa cell line KGN showed a high amount of non-vital cells after 48 h, the OvCar-3 cells of epithelial origin seemed to be more robust with respect to CIDCOD. To the best of our knowledge, we are the first to describe these ovarian cell type-specific differences with respect to CIDCOD. This finding might be especially relevant with respect to ovarian tissue cryopreservation and re-transplantation in fertility preservation, as ovarian tissue consists of cells of both endothelial and epithelial origin. An explanation for the observed differences in the viability of the OvCar-3 and KGN cells could be that the cell lines originated from different tumors. Whereas the origin of the KGN cell line was a granulosa cell tumor from the pelvic region, the OvCar-3 cell line was derived from an adenocarcinoma that had been resistant to cisplatin while sensitive to a number of other chemotherapeutic agents. KGN cells, in contrast to OvCar-3 cells, have aromatase activity and the ability to release estrogen and progesterone. We hypothesize that this might protect KGN cells from apoptosis, as the finding that estrogen can suppress apoptosis and promote tumor cell proliferation has already been demonstrated for several estrogen-dependent cancer cell types [27]. 

Moreover, the use of DMSO as a cryoprotective agent seems to be a contributing factor for cell survival, as reported earlier [26,28].

Baust and others demonstrated that controlling the cellular stress response not only improves cell survival, but also cell function and repopulation after warming [5,29]. This could reduce the need for cryoprotective agents such as DMSO [30,31]. Recently, it was shown by Liu et al. that a pre-vitrification treatment of ovarian grafts using rapamycin, an inhibitor of the mTOR signaling pathway, could effectively prevent apoptosis and promote ovarian survival during ovarian tissue cryopreservation [32]. If and how a rapamycin pretreatment could be helpful in preventing CIDCOD remains to be evaluated in further studies.

As mentioned above, it is still uncertain if a better outcome results from open or closed vitrification methods [19,20,33,34]. Despite the fact that our sample size appears too small to ultimately answer this question, our data support the assumption of equality between open and closed vitrification methods. When comparing the vitality of KGN and OvCar-3 cells using open and closed vitrification methods, we observed a significant difference between the two cell lines with respect to DMSO-containing and DMSO-free protocols. However, we could not find a significant difference between open and closed vitrification for the respective cell lines.

Of course, our study has a few limitations that should be taken into account. In our in vitro model we used two cell lines derived from ovarian tumors. Such cell lines are typically no longer subject to division restriction. Hence, our results may only be applicable to carcinoma cells. Furthermore, ovarian tissue consists of a multitude of different cell types, which influence each other through cell-to-cell contacts, exchange of growth factors, and endocrine crosstalk. In our in vitro situation, cells were cultivated only one cell type at a time. Therefore, our findings seem to be comparable to the in vivo tissue to only a limited extent. Including a non-cancerous ovarian cell line in future studies might help clarify if the results can be generalized to non-cancerous ovarian tissue.

## 4. Materials and Methods

The effects of vitrification were evaluated directly after warming via fluorescence-activated cell sorting (FACS) analysis, and, with respect to CIDOCD, 48 to 72 h after warming and cell cultivation. A granulosa cell line of endothelial origin (KGN) and an epithelial cell line (OvCar-3) were used as models for endothelial and epithelial cell types in ovarian tissue, respectively. The human KGN granulosa cell line was a friendly gift from Katja Horling (Department of Anatomy and Cell Biology, Faculty of Medicine, Martin Luther University, Halle, Germany). The OvCar-3 cell line was a friendly gift from Dan Cacsire Castillo-Tong (Department of Obstetrics and Gynecology, Medical University of Vienna, Vienna, Austria). Except otherwise stated, all chemicals were obtained from Sigma (Sigma Chemical Co., St. Louis, MO, USA).

A schematic illustration of the experimental schedule is presented in Figure 5. 

### 4.1. Vitrification and Warming

We adapted two published protocols: one described by Silber et al. [35] containing DMSO as the CPA, and one by Oktem et al. [36], who used a DMSO-free protocol. 

#### 4.1.1. DMSO-Containing Vitrification Protocol

For vitrification using the DMSO-containing protocol (DMSO-containing), 10 × 10^6^ cells were equilibrated for 5 min in a CPA solution containing 7.5% DMSO, 7.5% ethylene glycole (EG), and 20% fetal calf serum (FCS) in Dulbecco’s modified Eagle’s medium (DMEM). After centrifugation for 5 min at 300 rcf, the supernatant was discarded and the cells were incubated for 5 min in a solution containing 20% DMSO, 20% EG, 0.5 M sucrose, and 20% FCS in DMEM. After centrifugation, the supernatant was discarded and the cell pellets were cooled in a 1.8 mL tube (NUNC, Thermofisher Waltham, MA, USA) by direct immersion into liquid nitrogen (N_2_). For open vitrification (O), the tube was closed after the N_2_ step. For closed vitrification (CL), a lid was placed on the tube before immersion into liquid nitrogen. 

#### 4.1.2. DMSO-Free Vitrification Protocol

For vitrification using the DMSO-free protocol (DMSO-free), 10 × 10^6^ cells were equilibrated for 10 min in a CPA solution containing 15% propanediol, 15% EG, 0.2 M sucrose, and 10% FCS in DMEM. After centrifugation for 5 min at 300 rcf, the supernatant was discarded and the cell pellets were cooled in a 1.8 mL tube (NUNC, Thermofisher Waltham, MA, USA) by direct immersion into liquid nitrogen (N_2_). For open vitrification (O), the tube was closed after the N_2_ step. For closed vitrification (CL), a lid was placed on the tube before immersion into liquid nitrogen. 

#### 4.1.3. Warming

Warming of all samples was performed by adding 1 mL of a warming solution at 37 °C (containing 1 M sucrose and 20% FCS) for 1 min in a water bath at 37 °C. After centrifugation for 5 min at 300 rcf, the supernatant was discarded and 1 mL of a solution containing 0.5 M sucrose and 20% FCS was added to the cell pellet for 3 min. After a further centrifugation step, the cells were suspended in phosphate buffered saline (PBS) containing 20% FCS. 

### 4.2. Fluorescence-Activated Cell Sorting (FACS) Analysis 

For FACS analysis, vitrified tubes were warmed according to the protocol above. After two washing steps using PBS, cells were suspended in 1 mL PBS. Analyses were performed using a BD FACSVerse Flow cytometer. 4,6 diamino-2-phenylindole-dihydrochloride (DAPI) was added to the samples 10 min before the analysis started. DAPI negative cells were classified as vital. Experiments were repeated at least five times individually, and a minimum of 10,000 events were collected. Data were analyzed using BD FACSuite V1.06 and FLOWJO software (www.flowjo.com, accessed on 1 June 2023). 

### 4.3. Total Cell-Specific Vitality Rate

In the next step we calculated total cell-specific vitality rates depending on the protocol used (DMSO-containing vs. DMSO-free; open vs. closed vitrification) for all KGN cells and all OvCar-3 cells. 

We determined the percentage of cells that died during the cultivation period by subtracting the value after cultivation from the value immediately after warming, which reflects the percentage of cells that died within 48 h. We therefore subtracted the amount of vital cells 48 h after cultivation from the total amount of vital cells determined immediately after warming (FACS).

### 4.4. Cell Cultivation and Trypan Blue Staining 

Both cell lines, the human granulosa cell line, KGN, and the human ovarian carcinoma cell line, OvCar-3, were routinely cultured in DMEM medium supplemented with 10% fetal bovine serum (FBS) and a final concentration of 100 units/mL of penicillin and 100 μg/mL of streptomycin (1% Pen/Strep) at 37 °C with 5% CO_2._

Approximately 1.5 × 10^6^ KGN cells and 0.2 × 10^6^ OvCar-3 cells were seeded in cell culture dishes (Sarsted, Nürnberg, Germany). After 48 to 72 h the vitality of the cells was determined using trypan blue staining in a Neubaur’s counting chamber.

### 4.5. Statistical Analysis 

Data from FACS analyses and results of trypan blue staining were analyzed using ANOVA for independent samples followed by Tukey’s HSD Test on the Vassar Stat Homepage (www.Vassarstat.net). The rate of vital cells was handled as a numerical variable, and thus analysis of variances (ANOVA) was used to test this specific variable. ANOVA followed by a post hoc test is a frequently used statistical method to detect significances in FACS analyses dealing with apoptosis and necrosis. In our study, Tukey’s HSD Test for post hoc analysis was used [37,38]. Differences were considered statistically significant at *p* < 0.05. 

## 5. Conclusions

In conclusion, our results lend support to the hypothesis that cryopreservation-related damage is not only cell type dependent, but also dependent on the cryoprotective agent used. We believe that this observation requires further evaluation, especially in terms of ovarian tissue cryopreservation and subsequent re-transplantation.

## Figures and Tables

**Figure 1 ijms-24-12225-f001:**
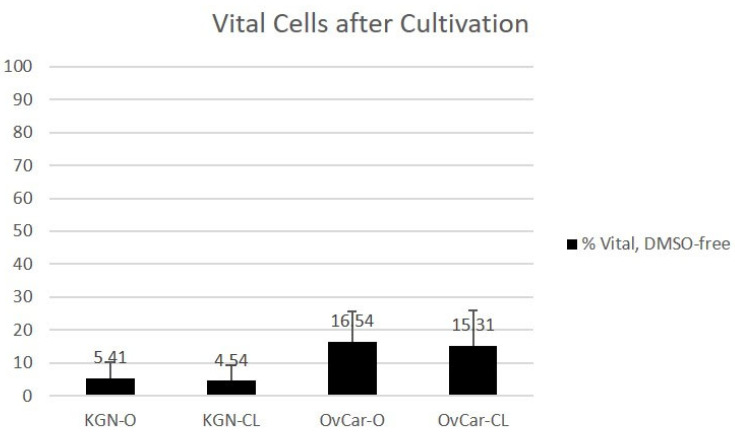
Cell vitality using a DMSO-containing protocol determined using FACS analysis. Results of FACS analyses of human granulosa cells (KGN) and ovarian cancer cells (OvCar) after open (O) or closed (CL) cryopreservation in DMSO-containing CPA solution. Individual experiments were repeated five times. Values are indicated in % ± SD.

**Figure 2 ijms-24-12225-f002:**
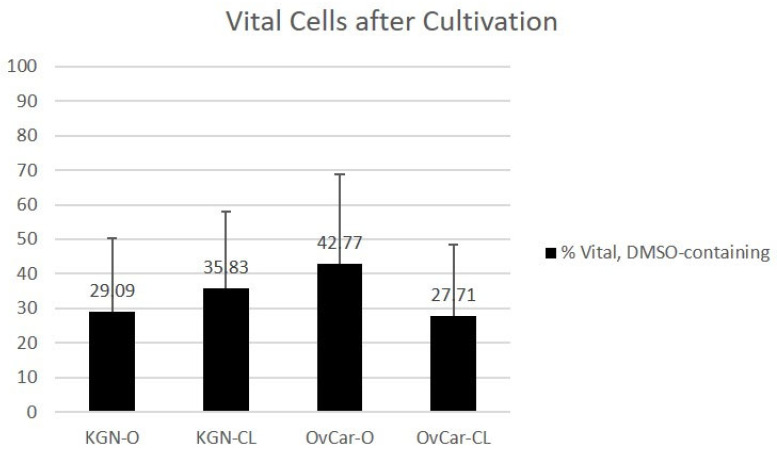
Cell vitality using a DMSO-free protocol determined using FACS analysis. Results of FACS analyses of human granulosa cells (KGN) and ovarian cancer cells (OvCar) after open (O) or closed (CL) cryopreservation in DMSO-free CPA solution. Individual experiments were repeated five times. Values are indicated in % ± SD.

**Figure 3 ijms-24-12225-f003:**
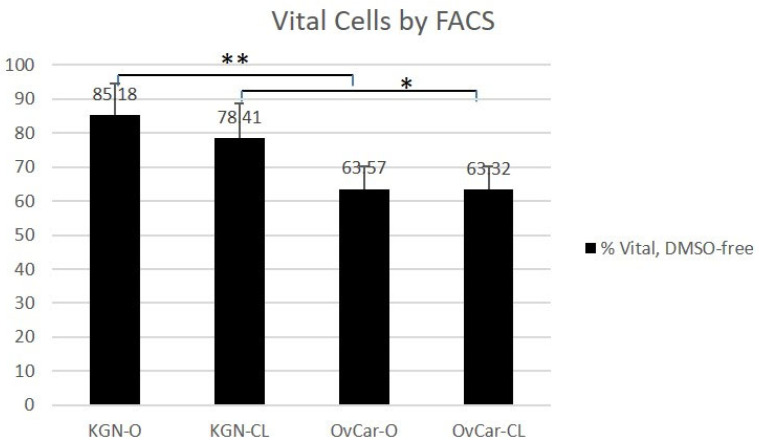
Cell vitality using a DMSO-containing protocol after cultivation and trypan blue staining. Results of trypan blue vitality counting of human granulosa cells (KGN) and ovarian cancer cells (OV) after open (O) or closed (CL) cryopreservation in DMSO-containing CPA solution and cultivation. (n = 5; values are indicated in % ± SD; significant parameters are marked with an asterisk (** significance level < 0.01, * significance level < 0.05).

**Figure 4 ijms-24-12225-f004:**
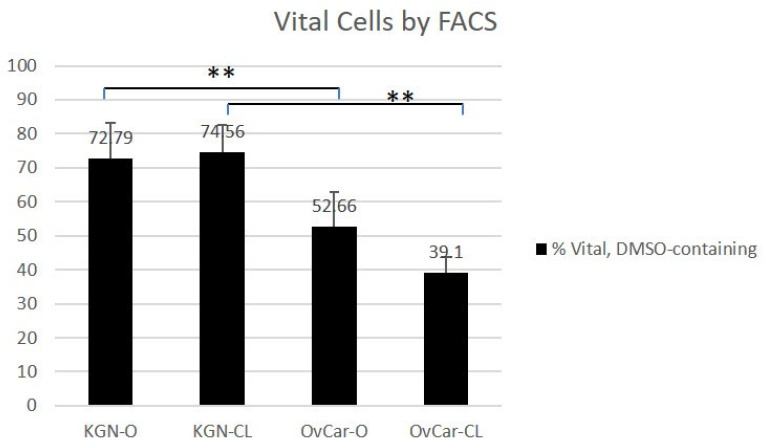
Cell vitality using a DMSO-free protocol after cultivation and trypan blue staining. Results of trypan blue vitality counting of human granulosa cells (KGN) and ovarian cancer cells (OV) after open (O) or closed (CL) cryopreservation in DMSO-free CPA solution and cultivation. (n = 5; values are indicated in % ± SD; significant parameters are marked with an asterisk; (** significance level < 0.01).

**Figure 5 ijms-24-12225-f005:**
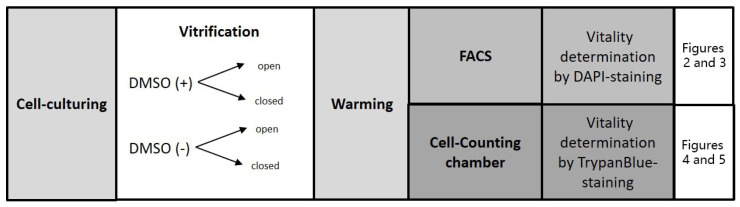
Experimental schedule for KGN and OvCar-3 cell lines. DMSO (+): DMSO-containing protocol; DMSO (−): DMSO-free protocol; FACS: fluorescence-activated cell sorting analysis; DAPI: 4,6 diamino-2-phenylindole-dihydrochloride.

**Table 1 ijms-24-12225-t001:** *Vitality rates*: percentage of non-viable cells immediately after warming and after 48 h of cultivation, sorted according to the CPAs used and open and closed vitrification.

Cell Line	DMSO Cont.	No DMSO	Open	Closed
KGN	41.22	76.82	61.74	56.3
OvCar	10.64	47.52	28.46	29.7

## Data Availability

Data will be made available upon request.

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
