# Peer review of "Different Impacts of Cryopreservation in Endothelial and Epithelial Ovarian Cells"

_ijms, 2023, doi:10.3390/ijms241512225_

Round 1

Reviewer 1 Report

Overall, the authors have undertaken a comprehensive investigation of vitrification methods for cryopreservation of ovarian cells, focusing particularly on delayed-onset cell death. They present the data in a clear manner and discuss the results in the context of the existing literature.

Some key points and areas for improvement are highlighted below:

1. The manuscript's strength lies in its exploration of a topic that is highly relevant to the field of cryopreservation and reproductive biology. The authors should be commended for focusing on cell death, an aspect often overlooked in cryopreservation research.

2. The use of both open and closed vitrification methods across two cell lines and with DMSO-containing and DMSO-free protocols has allowed for a thorough comparison. This comprehensive approach enhances the overall value of the study.

3. The use of ovarian cancer cell lines (KGN and OvCar-3) is a limitation. As the authors acknowledge, these cell lines may behave differently than normal ovarian cells due to their unrestricted division capabilities. Including a non-cancerous ovarian cell line might help to clarify if the results can be generalized to non-cancerous ovarian tissue.

4. The statistical analysis needs to be better explained. What specific tests were used, and why were these chosen? Providing this information would enhance the readers' ability to evaluate the data.

5. A more extensive discussion about the different sensitivities of the KGN and OvCar-3 cells to the vitrification processes would be beneficial. A deep-dive into known biological differences might provide insight into why these cell lines respond differently.

6. The authors suggest that DMSO improves cell survival, but it is unclear if other cryoprotectants were directly compared to DMSO. This needs to be clarified.

7. The introduction would benefit from a more thorough literature review. Providing more context and background would allow readers to better understand the significance of the research.

8. Minor formatting issues, such as incomplete references, should be addressed. 

In conclusion, this is a promising study that contributes valuable data to the field of cryopreservation. With some revisions, the manuscript could provide critical insights into optimizing ovarian tissue preservation.

The manuscript is generally well-written, with a clear structure and organization of the content. However, there are areas where grammar, syntax, and usage can be improved for better readability and precision. Below are some specific observations:

1. There are instances of awkward phrasing and word choice throughout the manuscript that could benefit from further editing. For example, "ultimatley" should be corrected to "ultimately". 

2. Some sentences are overly long and complex, which can disrupt the flow of the text and make it more difficult for the reader to follow. Consider breaking these into smaller, more manageable sentences. 

3. The use of hyphenation seems inconsistent and, in some cases, incorrect. For instance, the term "DMSO-con- taining" is hyphenated awkwardly due to line breaking, and "late onset cell death" might be more accurate as "late-onset cell death". 

4. Some abbreviations are not clearly defined before use (e.g., CIDCOD). Make sure all abbreviations are defined upon first use and used consistently thereafter.

5. The manuscript contains some typographical errors that need correction. For instance, the word "vitrifaction" should be corrected to "vitrification".

6. Certain terms or phrases could benefit from being more scientific and precise. For example, phrases like "loss of vital cells" might be better as "loss of cell viability".

7. The reference format appears to be inconsistent and incomplete. Please ensure to follow the chosen citation style accurately and consistently.

8. Lastly, ensure to maintain a formal, scientific tone throughout. For example, using "we were able to show" could be revised to "our results demonstrated".

By addressing these issues, the manuscript can be made more professional, accurate, and accessible to readers. However, these changes are relatively minor and do not detract significantly from the scientific quality of the study. I recommend a thorough proofread and language editing to polish the manuscript before submission or resubmission.

Author Response

Response to the editor and reviewers

We thank the editor and reviewers for their time and useful comments that have been addressed accordingly in the revised version of the manuscript. Please find a detailed response to each comment below.

Reviewer 1

  1. The use of ovarian cancer cell lines (KGN and OvCar-3) is a limitation. As the authors acknowledge, these cell lines may behave differently than normal ovarian cells due to their unrestricted division capabilities. Including a non-cancerous ovarian cell line might help to clarify if the results can be generalized to non-cancerous ovarian tissue.

Response: We totally agree with the reviewer. However, the KGN cell line is frequently used as it serves as the best cell line model for granulosa cell function due to its capability to produce progesterone and to a lower extend estrogen, resembling the function of luteinized human granulosa cells (Blanche 2022 und Nishi 2001).

The ovarian cancer cell line OvCar-3-3 was chosen for its responsiveness to various hormones by expression of typical ovarian hormone receptors e.g., androgen receptor, estrogen receptor and progesterone receptor (Hamilton 1983)

According to the reviewer’s suggestion, we now mention the possibility of an extension of our results to a more generalized cell line as a future research project in the discussion section of the manuscript.

--

3D Microtissues Mimic the Architecture, Estradiol Synthesis, and Gap Junction Intercellular Communication of the Avascular Granulosa

Blanche C Ip, Elizabeth Leary, Benjamin Knorlein, David Reich, Vivian Van, Joshua Manning, Jeffrey R Morgan. Toxicol Sci. 2022 Mar; 186(1): 29–42.

Establishment and characterization of a steroidogenic human granulosa-like tumor cell line, KGN, that expresses functional follicle-stimulating hormone receptor.

Nishi Y, Yanase T, Mu Y, Oba K, Ichino I, Saito M, Nomura M, Mukasa C, Okabe T, Goto K, Takayanagi R, Kashimura Y, Haji M, Nawata H. Endocrinology. 2001 Jan;142(1):437-45. doi: 10.1210/endo.142.1.7862.

Characterization of a human ovarian carcinoma cell line (NIH:OVCAR-3) with androgen and estrogen receptors Cancer Res . 1983 Nov;43(11):5379-89.
T C HamiltonR C YoungW M McKoyK R GrotzingerJ A GreenE W ChuJ Whang-PengA M RoganW R GreenR F Ozols

  1. The statistical analysis needs to be better explained. What specific tests were used, and why were these chosen? Providing this information would enhance the readers' ability to evaluate the data.

Response: We thank the reviewer for his/her comment and added the following para in the Materials and Methods Section of the manuscript:

“The data of the FACS analyses and the results of the Trypan blue staining were analyzed using ANOVA for independent samples followed by Tukey’s HSD Test on the Vassar Stat Homepage (www.Vassarstat.net). The rate of vital cells was handled as a numerical variable and thus the analysis of variances (ANOVA) was used to test this specific variable. ANOVA, followed by post hoc test is a frequently used statistical method to detect significances in FACS analyses dealing with apoptosis and necrosis. In our study Tukey’s HSD Test for post hoc analysis was used (Lew 2007 und Kenmotsu 2010). Differences were considered statistically significant at p < 0.05.”

--

Good statistical practice in pharmacology. Problem 2.
Lew M. Br J Pharmacol. 2007 Oct;152(3):299-303. doi: 10.1038/sj.bjp.0707372. Epub 2007 Jul 9. PMID: 17618310

Analysis of side population cells derived from dental pulp tissue.
Kenmotsu M, Matsuzaka K, Kokubu E, Azuma T, Inoue T. Int Endod J. 2010 Dec;43(12):1132-42. doi: 10.1111/j.1365-2591.2010.01789.

  1. A more extensive discussion about the different sensitivities of the KGN and OvCar-3 cells to the vitrification processes would be beneficial. A deep-dive into known biological differences might provide insight into why these cell lines respond differently.

Response: We thank the reviewer for the comment and changed respective para in the discussion section of the manuscript:

“An explanation for the observed differences in the viability of the OvCar-3 and KGN cells could be that the cell lines originated from different tumors. While the origin of the KGN cell line was a granulosa cell tumor from the pelvic region, the OvCar-3 cell line was derived from an adenocarcinoma that had been resistant to cisplatin while sensitive to a number of other chemotherapeutic agents. KGN cells, in contrast to OvCar-3 cells, have aromatase activity and the ability to release estrogen and progesterone. We hypothesize that this might protect KGN cells from apoptosis, as the finding that estrogen can suppress apoptosis and promote tumor cell proliferation has already been demonstrated for several estrogen-dependent cancer cell types (Chimento 2012).”

--

The putative G-protein coupled estrogen receptor agonist G-1 suppresses proliferation of ovarian and breast cancer cells in a GPER-independent manner
Cheng Wang,1,2 Xiangmin Lv,1,2,4 Chao Jiang,1,2 and John S Davis1,2,3
Am J Transl Res. 2012; 4(4): 390–402.

Estrogen Receptors-Mediated Apoptosis in Hormone-Dependent Cancers
Adele Chimento  1 , Arianna De Luca  1 , Paola Avena  1 , Francesca De Amicis  1 , Ivan Casaburi  1 , Rosa Sirianni  1 , Vincenzo Pezzi  1
Int J Mol Sci 2022 Jan 22;23(3):1242. doi: 10.3390/ijms23031242.

Estrogen inhibits paclitaxel-induced apoptosis via the phosphorylation of apoptosis signal-regulating kinase 1 in human ovarian cancer cell lines
Seiji Mabuchi  1 , Masahide Ohmichi, Akiko Kimura, Yukihiro Nishio, Emi Arimoto-Ishida, Namiko Yada-Hashimoto, Keiichi Tasaka, Yuji Murata
Endocrinology . 2004 Jan;145(1):49-58. doi: 10.1210/en.2003-0792

Molecular mechanisms of estrogen action: selective ligands and receptor pharmacology. Katzenellenbogen BS, Choi I, Delage-Mourroux R, Ediger TR, Martini PG, Montano M, Sun J, Weis K, Katzenellenbogen JA.
J Steroid Biochem Mol Biol. 2000 Nov 30;74(5):279-85. doi: 10.1016/s0960-0760(00)00104-7.

  1. The authors suggest that DMSO improves cell survival, but it is unclear if other cryoprotectants were directly compared to DMSO. This needs to be clarified.

Response: We thank the reviewer for this comment. Published data of our study group revealed that in ovarian tissue vitrification a protocol containing DMSO results in a higher viability of ovarian cells than a protocol that uses ethylene glycol as cryoprotective agent (Marschalek et al. 2021). In addition, we were able to demonstrate a beneficial effect of DMSO as a cryoprotectant on granulosa cell vitrification compared to ethylene glycol (Kokotsaki et al 2018). We now address this in the discussion section of our manuscript.

--

The effect of different vitrification protocols on cell survival in human ovarian tissue: a pilot study.
Marschalek J, Egarter C, Nouri K, Dekan S, Ott J, Frank M, Pietrowski D.
J Ovarian Res. 2021 Dec 6;14(1):170. doi: 10.1186/s13048-021-00924-8.

Impact of vitrification on granulosa cell survival and gene expression.
Kokotsaki M, Mairhofer M, Schneeberger C, Marschalek J, Pietrowski D.
Cryobiology. 2018 Dec;85:73-78. doi: 10.1016/j.cryobiol.2018.09.006. Epub 2018 Sep 25.

  1. The introduction would benefit from a more thorough literature review. Providing more context and background would allow readers to better understand the significance of the research.

Response: We adapted the introduction section of the manuscript according to the reviewer’s suggestion.

  1. Minor formatting issues, such as incomplete references, should be addressed. 

Response: corrected.

  1. There are instances of awkward phrasing and word choice throughout the manuscript that could benefit from further editing. For example, "ultimatley" should be corrected to "ultimately". 

Response: corrected

  1. Some sentences are overly long and complex, which can disrupt the flow of the text and make it more difficult for the reader to follow. Consider breaking these into smaller, more manageable sentences. 

Response: We thank the reviewer for this comment and rephrased respective sentences.

  1. The use of hyphenation seems inconsistent and, in some cases, incorrect. For instance, the term "DMSO-con- taining" is hyphenated awkwardly due to line breaking, and "late onset cell death" might be more accurate as "late-onset cell death".

Response: corrected

  1. Some abbreviations are not clearly defined before use (e.g., CIDCOD). Make sure all abbreviations are defined upon first use and used consistently thereafter.

Response: CIDOCD is defined in the 3rd paragraph of the introduction section. This is the first section containing this abbreviation.

  1. The manuscript contains some typographical errors that need correction. For instance, the word "vitrifaction" should be corrected to "vitrification".

Response: corrected

  1. Certain terms or phrases could benefit from being more scientific and precise. For example, phrases like "loss of vital cells" might be better as "loss of cell viability".

Response: corrected

  1. The reference format appears to be inconsistent and incomplete. Please ensure to follow the chosen citation style accurately and consistently.

Response: The reference list has been adapted.

  1. Lastly, ensure to maintain a formal, scientific tone throughout. For example, using "we were able to show" could be revised to "our results demonstrated".

Response: We adapted respective wording according to the reviewer’s suggestion.

Reviewer 2 Report

The title of manuscript is remarkable. English language has good quality. Figures and Tables have acceptable quality. Main text need some modifications. Some sentences also need proper reference. Citations need some modifications.

1. In page 1, section "Abstract", part "Background"

In this part the authors have written some brief data about the aim and the protocol by which they have performed their survey. Please rewrite this part and transfere the sentences of this part to their proper part in this section

2. All multipple and middle sentence references in all over the manuscript should be reformed

3. In page 2, line "Thus, evaluating the effectivity of a cryopreservation ... the times

needed for the respective equilibration steps"

The sentences of this part needs proper references

4. About the section "Discussion"

Please categorize your results based on their importance from the most important one to the least. After that, discuss about each one of them one by one.

5. Please check and adjust the "Reference list" based on the regulations of reference list of journal. (Titles, doi, the name of journal and ... )

Author Response

Response to the editor and reviewers

We thank the editor and reviewers for their time and useful comments that have been addressed accordingly in the revised version of the manuscript. Please find a detailed response to each comment below.

Reviewer 2

A few questions remaining to be answered are if the use of open or closed vitrification devices make a difference concerning the vitrification of specific cell types and whether late-onset cell death can also occur in vitrified ovarian tissue. If this was the case, it would be interesting to know which cell type is the most affected

From the existing literature on this topic, it is not fully cleared if open or closed vitrification devices are the better choice in the vitrification process of specific cell types.

Response: We thank the reviewer for these comments and totally agree, that existing literature has not fully answered all the risen questions about ovarian tissue vitrification yet. Our study group focused on this topic in the past years and concluded that a protocol containing DMSO results in a higher viability of ovarian cells than a protocol that uses ethylene glycol as cryoprotective agent in vitrification of ovarian tissue. Moreover, the use of an open vitrification system led to a significant decline in the rate of viable cells. Which cell type in subsequent warming is most affected by late onset cell death has not sufficiently been answered so far.

--

Co-cultivation of human granulosa cells with ovarian cancer cells leads to a significant increase in progesterone production.
Pietrowski D, Grgic M, Haslinger I, Marschalek J, Schneeberger C.Arch Gynecol Obstet. 2023 May;307(5):1593-1597. doi: 10.1007/s00404-023-06914-z. Epub 2023 Jan 18.

The effect of different vitrification protocols on cell survival in human ovarian tissue: a pilot study.
Marschalek J, Egarter C, Nouri K, Dekan S, Ott J, Frank M, Pietrowski D.J Ovarian Res. 2021 Dec 6;14(1):170. doi: 10.1186/s13048-021-00924-8.

  1. In page 1, section "Abstract", part "Background"

In this part the authors have written some brief data about the aim and the protocol by which they have performed their survey. Please rewrite this part and transfere the sentences of this part to their proper part in this section

Response: We adapted the introduction section of the manuscript according to the reviewer’s suggestion.

  1. 2. All multipple and middle sentence references in all over the manuscript should be reformed

Response: We thank the reviewer for this comment and rephrased respective sentences.

  1. 3. In page 2, line "Thus, evaluating the effectivity of a cryopreservation ... the times

needed for the respective equilibration steps"

The sentences of this part needs proper references

Response: Thank you - we adapted the references according to the reviewer’s suggestion.

  1. About the section "Discussion"

Please categorize your results based on their importance from the most important one to the least. After that, discuss about each one of them one by one.

Response: We adapted the discussion section of the manuscript as wished by Reviewer 2 by discussing our results from the most important finding to the last.

  1. Please check and adjust the "Reference list" based on the regulations of reference list of journal. (Titles, doi, the name of journal and ... )

Response: The reference list has been adapted.

Round 2

Reviewer 1 Report

Accept in current form